# Insights into Reactive Oxygen Species Production-Scavenging System Involved in Sugarcane Response to *Xanthomonas albilineans* Infection under Drought Stress

**DOI:** 10.3390/plants13060862

**Published:** 2024-03-17

**Authors:** Yao-Sheng Wei, Jian-Ying Zhao, Talha Javed, Ahmad Ali, Mei-Ting Huang, Hua-Ying Fu, Hui-Li Zhang, San-Ji Gao

**Affiliations:** 1National Engineering Research Center for Sugarcane, Fujian Agriculture and Forestry University, Fuzhou 350002, China; dell990218@163.com (Y.-S.W.); zhaojyfafu@126.com (J.-Y.Z.); Ahmad03348454473@yahoo.com (A.A.); hmt159379@163.com (M.-T.H.); mddzyfhy@163.com (H.-Y.F.); hlzhang@fafu.edu.cn (H.-L.Z.); 2Institute of Tropical Bioscience and Biotechnology, Chinese Academy of Tropical Agricultural Sciences, Haikou 571101, China; talhajaved@itbb.org.cn

**Keywords:** *Saccharum* spp. hybrids, reactive oxygen species, leaf scald, drought, defense response

## Abstract

Plants must adapt to the complex effects of several stressors brought on by global warming, which may result in interaction and superposition effects between diverse stressors. Few reports are available on how drought stress affects *Xanthomonas albilineans* (*Xa*) infection in sugarcane (*Saccharum* spp. hybrids). Drought and leaf scald resistance were identified on 16 sugarcane cultivars using *Xa* inoculation and soil drought treatments, respectively. Subsequently, four cultivars contrasting to drought and leaf scald resistance were used to explore the mechanisms of drought affecting *Xa*–sugarcane interaction. Drought stress significantly increased the occurrence of leaf scald and *Xa* populations in susceptible cultivars but had no obvious effect on resistant cultivars. The ROS bursting and scavenging system was significantly activated in sugarcane in the process of *Xa* infection, particularly in the resistant cultivars. Compared with *Xa* infection alone, defense response via the ROS generating and scavenging system was obviously weakened in sugarcane (especially in susceptible cultivars) under *Xa* infection plus drought stress. Collectively, ROS might play a crucial role involving sugarcane defense against combined effects of *Xa* infection and drought stress.

## 1. Introduction

The number and frequency of extreme weather events have increased significantly due to global climate change and pose a serious impact on the sustainable development of agriculture [1,2]. Recent research has shown that extreme environmental events have a significant impact on the pathogenicity and transmissibility of pathogenic microorganisms [3], which will impair plant disease resistance, alter defensive signaling pathways, and ultimately cause a dramatic decline in plant growth and survival [4,5]. Thus, it is crucial to comprehend how abiotic stresses affect biotic stresses in order to engineer plants for climate adaptation and to ensure the long-term viability of agriculture [3].

Crosstalk and trade-offs exist in plants in response to abiotic and biotic stresses [3]. Drought directly weakens the fitness and survival of a wide range of root and leaf pathogens that require moisture [6]. Furthermore, combined drought and pathogen stress alters physio-morphological traits such as photosynthesis, stomatal conductance, and transpiration rate along with plant growth and root morphology [7]. However, some cases reported that plants which are subjected to drought stress increase plant susceptibility by destroying the woody structure, resulting in the acceleration of some diseases, such as charcoal stalk rot in sorghum and smut in cereals [3].

Reactive oxygen species (ROS) burst is an effective approach for plants to fend off stressors in the early stages of unfavorable environmental conditions. Later, ROS act as signaling molecules that set off the organism’s defensive systems [8,9]. However, the plant loses its ability to respond to osmotic stress when the ROS concentrations surpass the threshold value; in extreme circumstances, this may result in wilting or even death [10,11]. ROS homeostasis in plants mainly includes enzymatic and non-enzymatic scavenging systems [12]. The enzymatic scavenging system mainly includes superoxide dismutase (SOD), catalase (CAT), peroxidase (POD), ascorbate peroxidase (APX), and so on [12]. Among them, SOD is the first and most important line of defense in the enzymatic scavenging system due its ability to remove superoxide ($O_{2}^{-}$) and generate the disproportionation product H_2_O_2_ [10]. In response to pathogen infection, plants trigger hypersensitivity reactions and apoptosis through ROS burst, thereby limiting the proliferation of pathogens [13,14]. In addition, drought confers resistance to plants against subsequent pathogen infections by elevated levels of ROS generation and higher activity of the ROS scavenging system [7].

Sugarcane (*Saccharum* spp. hybrids), a classic C_4_ crop with the highest photosynthetic rates among other crops, plays a crucial role in sugar and biofuel production, thereby accounting for 80% of sugar production in the world and approximately 90% in China. China’s sugarcane-producing areas are mainly located in the arid slope land of southern and southwestern regions such as Guangxi and Yunnan provinces, where sugarcane yield loss often occurs due to poor irrigation facilities and soil water retention capacity coupled with uneven natural precipitation. Extreme drought can lead to a reduction of up to 60% in sugarcane yields [15]. The bacterial pathogen *Xanthomonas albilineans* (*Xa*) is a causal agent of leaf scald in sugarcane, distributed across the majority of sugarcane-planting counties worldwide [16]. However, the mechanism of sugarcane infection by *Xa* under drought stress remains unclear. This study aims to assess some newly released sugarcane cultivars resistant to drought stress and *Xa* infection. Additionally, the ROS production-scavenging system participating sugarcane response to both stressors has been illustrated.

## 2. Results

### 2.1. Identification of Sugarcane Cultivars Tolerant to Drought

The malondialdehyde (MDA) contents in the leaves of 16 sugarcane cultivars were measured under drought stress. The MDA contents of all sugarcane cultivars increased significantly after drought stress. Notably, the MDA contents of ROC22, GT16-1253, and LC09-15 were significantly lower than that of other cultivars. The leaf relative water contents and maximum quantum yield of PSII photochemistry (Fv/Fm) of all sugarcane cultivars decreased after drought stress. The non-photochemical quenching (qN) of FN04-3504, GT29, YG59, ZZ14, and ZZ13 cultivars was significantly decreased after drought stress. Following drought stress, the PSII real photosynthetic efficiency (Y(II)) suffered a significant drop in FN04-3504, GT29, YG59, YZ15-505, and ZZ13 (Figure 1).

The membership function values and comprehensive evaluation values (D values) of 16 sugarcane cultivars were calculated by using the membership function method of fuzzy mathematics (Table 1). As shown in Table 1, the D value was positively correlated with drought resistance. Among all the tested cultivars, LC09-15 had the highest D value, while GT29 had the smallest. Finally, the drought resistance of the 16 sugarcane cultivars at the seedling stage was graded from strong to weak (Table 1).

### 2.2. Identification of Sugarcane Cultivars Resistant to Leaf Scald

The disease index was calculated according to the severity classification scale of leaf scald among 16 sugarcane cultivars. After Xa-FJ1 inoculation, the disease index varied for each variety of sugarcane but rose as the inoculation period was extended cultivars. Among them, ZZ13 had the lowest disease index (11.8%) after inoculation with the Xa-FJ1 strain, while ZZ14 had the highest disease index (58.8%). These findings demonstrated that the levels of resistance to the leaf scald varied significantly in these tested cultivars (Figure 2).

Based on the disease index 28 days post-inoculation (dpi) with the Xa-FJ1 strain, the resistance of 16 sugarcane cultivars to leaf scald disease was divided into four grades, as shown in Table 2. The disease index of the resistant group, which comprised ROC22 and ZZ13, varied between 11.8% and 12.6%. The disease index of the medium resistant group, including FN14-1854, LC05-136, and YZ15-505, ranged from 15.1% to 26.3%. The susceptible group consisted of seven cultivars (i.e., YG59, ZZ8, etc.), and the disease index ranged from 29.4% to 48.4%. There were four cultivars (GT29, ZZ14, LC09-15, and ZT1501) in the high susceptible group, and the disease index ranged from 52.8% to 58.8%.

Based on the results of the identification of the drought and leaf scald resistance of 16 sugarcane cultivars, 4 cultivars were selected for follow-up experiments: ROC22 (disease-resistant and drought-tolerant), ZZ13 (disease-resistant and non-drought-tolerant), GT29 (susceptible and non-drought-tolerant), and LC09-15 (susceptible and drought-tolerant).

### 2.3. Effect of Drought Stress on the Occurrence of Leaf Scald in Sugarcane

To explore the effect of drought stress on the infection of sugarcane by *Xa*, four sugarcane cultivars, ROC22, ZZ13, GT29, and LC09-15, were treated with Xa-FJ1 infection alone and combined stress (Xa-FJ1 infection plus PEG6000 osmotic stress, Xa + PEG6000). The bacterial contents of all the sugarcane cultivars increased significantly after two treatments (Figure 3). At 24 h post-treatment (hpt), the pathogenic bacterium populations in two cultivars (GT29 and LC09-15) susceptible to leaf scald were significantly increased by 143.1% and 39.5% in combined stress compared with Xa-FJ1 infection alone, respectively. Namely, the bacterial populations of GT29 and LC09-15 reached 8507.89 copies/L and 7885.44 copies/L under combined stress, respectively. By contrast, the bacterial populations of both cultivars were 3499.70 copies/L and 5651.30 copies/L under Xa-FJ1 infection alone, respectively. On the other hand, the bacterial populations of two cultivars (ROC22 and ZZ13) resistant to leaf scald did not have significant differences between both treatments at 24 pht. Overall, the proliferation rate of Xa-FJ1 in susceptible cultivars increased significantly under drought stress, but there was no significant difference in the proliferation rate of this pathogenic bacterium in resistant cultivars.

### 2.4. Changes in ROS Contents and ScRBOHD Gene Expression under Xa Infection Plus Drought Stress

The changes in the ROS contents in leaves and the transcriptional expression of *ScRBOHD* (a key gene for ROS synthesis) in four sugarcane cultivars under different treatments are shown in Figure 4. After Xa-FJ1 inoculation alone, the ROS contents in the leaves of ROC22, ZZ13, GT29, and LC09-15 were increased by 1.7-, 1.9-, 1.3-, and 1.4-fold, respectively, compared to the control (0 hpi). Correspondingly, the expression levels of the *ScRBOHD* gene rose by 7.5-, 5.0-, 3.1-, and 2.4-fold at 24 hpt compared to those of the control (0 hpi), respectively. Under combined stress, the ROS contents and *ScRBOHD* expression levels in four sugarcane cultivars were significantly increased, i.e., compared with 0 hpt, the ROS contents and *ScRBOHD* expression levels were enhanced, with increases of 1.2–1.8-fold and 1.6–8.0-fold in four cultivars at 24 hpt, respectively. On the other hand, the ROS contents and *ScRBOHD* expression levels in the two GT29 and LC09-15 cultivars were significantly lower at 24 hpt under combined stress than those under Xa-FJ1 inoculation alone. Namely, the ROS contents and *ScRBOHD* expression levels in the two GT29 and LC09-15 cultivars were decreased by 9–13% and 27–33% at 24 hpt under combined stress, respectively, compared to *Xa* inoculation alone. Meanwhile, no significant differences in the ROS contents and *ScRBOHD* expression levels were found at 24 hpt between Xa-FJ1 inoculation alone and combined stress in two ROC22 and ZZ13 cultivars.

### 2.5. Changes in Antioxidant Enzyme Activity and Gene-Related Expression under Xa Infection Plus Drought Stress

The activities of two key antioxidant enzymes (SOD and CAT) and the transcriptional expression of their related genes were measured in four sugarcane cultivars under different treatments (Figure 5). Under Xa-FJ1 inoculation alone, the SOD activities in four tested cultivars were increased by 9–12% at 24 hpi, while the expression levels of the *ScSOD* gene were enhanced 0.6–1.9-fold at 24 hpt compared with 0 hpi. Under combined stress, the SOD activities and expression levels of *ScSOD* in the four cultivars were increased. Namely, SOD activities were increased 8–9% and *ScSOD* expression levels were increased 1.4–2.0-fold in both cultivars ROC22 and ZZ13 at 24 hpt compared with 0 hpt. Meanwhile, the SOD activities and *ScSOD* expression levels in GT29 and LC09-15 cultivars at 24 hpt were significantly lower than those of Xa-FJ1 inoculation alone. However, no significant difference in the SOD activity and *ScSOD* expression was observed between combined stress and Xa-FJ1 infection alone in the two cultivars ROC22 and ZZ13. 

Under *Xa* infection alone, CAT activities decreased, ranging from 11% to 31%, and the expression levels of *ScCAT* decreased from 20% to 44% in the four tested cultivars at 24 hpt compared with 0 hpt. Under combined stress, CAT activities were decreased by 12–32% in three cultivars (ROC22, ZZ13, and GT29), while the expression levels of *ScCAT* in ROC22 and ZZ13 were decreased by 30–50% at 24 hpt compared with 0 hpt. Overall, the CAT activity and expression level of *ScCAT* in the four cultivars were decreased or not significantly changed under *Xa* infection alone or combined stress. No significant difference in CAT activity or *ScCAT* expression level was found between *Xa* infection alone and combined stress in the four cultivars.

The frequency of extreme weather events has increased due to global warming, which has had a significant effect on sugarcane yield and quality [1,14]. Drought has emerged as a predominant factor restricting sugarcane production in China [17]. Numerous studies have highlighted the importance of relative water content, MDA, and chlorophyll fluorescence as pivotal indicators for assessing plant drought resistance [18,19]. Additionally, some parameters such as gauge hydration status, oxidative damage, and photosynthetic efficiency provide comprehensive insights into plant responses to drought stress [7]. In this study, a diverse array of indicators and statistic methods for assessing drought resistance were measured, including these key parameters, to comprehensively evaluate plant responses to drought stress. Some drought-tolerant cultivars such as LC09-15 and GT16-1253 were proposed. In addition, exploring and utilizing disease-resistant germplasm resources has become the most cost-effective and efficient approach for preventing and controlling sugarcane diseases [20,21]. Thus, some cultivars such as ROC22 and ZZ13 resistant to leaf scald were identified. These findings suggested that some sugarcane cultivars like ROC22 are tolerate to drought and leaf scald. To our knowledge, the cultivar ROC22 has even occupied 85% of sugarcane plant areas during the years of 2005–2021, and now it is becoming the main hybrid parent for sugarcane genetic improvement in China.

Abiotic and biotic stressors have together been demonstrated to alter plant defense signaling pathways, as well as the pathogenicity of pathogens, which in turn has a more detrimental effect on agricultural productivity [22,23]. Drought can affect the severity of plant diseases [7]. For instance, drought stress reduces the severity of diseases in *Nicotiana benthamiana* infected by *Sclerotinia sclerotiorum* and *Pseudomonas syringae* pv. *tabaci* [24]. Furthermore, drought-induced stomatal closure restricts the pathogens of *Pseudomonas syringae* and *Melampsora apocyni* infecting *Arabidopsis thaliana* and *Apocynum venetum*, respectively, consequently diminishing disease incidence [25,26]. On the contrary, certain diseases such as charcoal stalk rot in sorghum, smut in cereals, and dry root rot in chickpeas exhibited rapid escalation under drought conditions, which might be caused by the fact that drought exacerbates plant vulnerability by compromising the structural integrity of plant tissues, especially the woody components [6,27,28]. In addition to impacting plants, drought directly hampers the growth and spread of pathogenic bacteria, thereby influencing their pathogenesis. For instance, drought stress significantly restricts the movement of *Verticillium dahliae* conidia, leading to a notable reduction in wilt severity [29]. Plants subjected to severe drought stress are not of benefit to pathogens with a biotrophic phase [7]. Our investigations showed that sugarcane exposed to PEG6000 stress had considerably higher populations of *Xa* in vulnerable cultivars, indicating that drought stress may enhance the incidence of sugarcane leaf scald.

Plants respond to stress by producing a wide range of intricate and sophisticated defense signals [30,31]. Notably, one of the key defense mechanisms of plants is ROS burst [8,10]. The respiratory burst oxidase homolog encoded by the *RBOH* gene is a key enzyme that produces ROS in response to stress signaling [32]. In this study, the levels of ROS production and *ScRBOHD* expression in sugarcane plants, especially in resistant cultivars, showed a significant increase after *Xa* infection. These results suggest that ROS is involved in the process of *Xa* infection, and the intensity of ROS burst is potentially correlated with the tolerance to leaf scald in sugarcane. Previous studies also illustrated that ROS production and *ScRBOHD* expression were involved in sugarcane in response to *Xa* stimuli [33] in rice cultivars resistant to *Magnaporthe oryzae* [34] and in *Gossypium barbadense* resistant to *Verticillium dahliae* [35]. On the other hand, the SOD and CAT are important protective enzymes in the enzymatic scavenging system of ROS, participating in plant defense responses under adverse environmental conditions [8,10]. Our study showed that SOD activity and *ScSOD* gene expression were markedly increased, while CAT activity and *ScCAT* gene expression were decreased in sugarcane, particularly in resistant cultivars, following *Xa* infection. This phenomenon could be attributed to the excessive accumulation of ROS in plants to induce sugarcane possessing stronger defense responses under the infection of this pathogen. Another obversion suggested that this phenomenon could be attributed to the excessive accumulation of salicylic acid in plants under pathogen stress, leading to a reduction in CAT activity [36].

Numerous investigations have demonstrated that a crucial node in the interplay between biotic and abiotic stressors is the ROS burst and scavenging mechanism [22,37]. ROS usually resists a combination of biotic and abiotic stresses by modulating ABA signaling and enhancing disease resistance [3,23]. Some cases revealed that drought imparts tolerance to plants against subsequent pathogen infections by enhanced levels of ROS and higher activity of key antioxidant enzymes [7,24]. For example, the plant of *N. benthamiana* exposure to moderate drought stress resulted in higher ROS production and the induction of some defense genes, decreasing the severity of subsequent infection by *P. syringae* pv. *tabaci* [24]. On the contrary, the rice plant subjected to drought stress increased the susceptibility to *M. oryzae* through downregulating the expression of multiple resistance genes and a reduction in ROS production [38]. Notably, our results showed that the levels of ROS and SOD activity in susceptible cultivars exposed to PEG6000 stress combined with *Xa* infection were markedly higher than those under *Xa* infection alone. A possible reason for increased ROS production might be drought stress. There are many studies that report increased ROS accumulation and oxidative stress under drought stress [39,40,41]. 

## 3. Materials and Methods

### 3.1. Plant Growth of Sugarcane

Healthy stalks of 16 sugarcane cultivars were collected from a sugarcane nursery (Fuzhou, China), and the single cuttings were immersed in flowing tap water overnight. These cuttings were then immersed in hot water (52 °C) for 3 h. The treated buds were transplanted into a 6 cm^3^ planting pot filled with nutrient soil and subsequently placed in an intelligent artificial climate chamber (PLT-RGS-15PF, Ningbo Prandt Instrument Co., Ltd., Ningbo, China) with the following culture conditions: 28 °C, 65% humidity, a light/dark cycle of 16/8 h, and a light intensity of 30,000 Lux. Soil moisture was consistently maintained at 80% through artificial management. After 25 days, sugarcane plants grown until they reached the stage of 3–5 leaves (15–20 cm tall) were used for drought treatment and/or *Xa* inoculation. The soil moisture contents in pots were controlled at 80% before treatment. 

### 3.2. Soil Drought Treatment

A total of 30 buds of sugarcane were used, of which 15 buds were used for soil drought treatment, and the other 15 buds within a controlled environment (80% soil moisture) were used as the control. Three independent repeats were carried out for each experiment. Each experiment was conducted using the Randomized Complete Block Design (RCBD) for allocation. This experiment was carried out in an intelligent artificial climate chamber with the abovementioned conditions. Leaf samples were collected at 0 and 3 days after drought treatment for the subsequent determination of drought resistance indicators. 

### 3.3. Determination of the Drought Indexes

The following formula for calculating the leaf relative water contents (RWCs) of sugarcane was used:$$\mathrm{RWC}=\left(\mathrm{fresh}\;\mathrm{weight}-\mathrm{dry}\;\mathrm{weight}\right)÷\left(\mathrm{turgid}\;\mathrm{weight}-\;\mathrm{dry}\;\mathrm{weight}\right)\times 100$$

The maximum quantum yield of PSII photochemistry (Fv/Fm), non-photochemical quenching (qN), and PSII actual photosynthetic efficiency (Y(Ⅱ)) was measured by the Imaging-PAM Chlorophyll Fluorometer (Shanghai Zealquest Scientific Technology Co., Ltd., China). Sugarcane plants were kept for 30 min in the dark before measurement. 

The MDA content of the leaves from treated plants was analyzed using the kit of the biological company, and the specific experimental process was performed according to the manufacturer’s manual (Beijing Solarbio Science & Technology Co., Ltd., China). Briefly, we weighed about 0.1 g of leaves and added 1 mL of extract buffer, and then they were ground for homogenization under an ice bath condition. They were centrifuged at 8000× *g* at 4 °C for 10 min, and then the supernatant was taken. We added reagents according to the instructions and kept the mixture in a 100 °C water bath for 60 min, and subsequently it was placed in an ice bath to cool. Then it was centrifuged at 10,000× *g* at room temperature for 10 min. The absorbance of the supernatant was measured at 532 nm and 600 nm wavelength using a Synergy H1 Multi-Mode Reader (BioTek, Winooski, VT, USA). 

### 3.4. Analysis of Drought-Resistance Grades of Sugarcane Cultivars

The drought-resistance grades of sugarcane cultivars were calculated based on the physiological indicators determined after drought stress using the membership function method in fuzzy mathematics. The calculation method used is as follows.

If the measured index is positively correlated with drought resistance,
$$\left(X_{i}\right)=\left(X_{i}-X_{min}\right)÷\left(X_{max}-X_{min}\right)$$
if the measured index is negatively correlated with drought resistance,
$$R\left(X_{i}\right)=1-\left(X_{i}-X_{min}\right)÷\left(X_{max}-X_{min}\right)$$

The letters in the formula indicate that *X_i_* = the ratio of the drought resistance index between treatment and control groups; *X_min_* and *X_max_* stand for minimum and maximum values of *X_i_* in the measured indexes among all the tested cultivars, respectively. 

### 3.5. X. albilineans Inoculation

To assess the disease resistance of the tested sugarcane cultivars, the decapitation method was used following the method of Zhao et al. [33]. The Xa-FJ1 strain of *X. albilineans* was used for inoculation [42]. A set of 35 buds of each sugarcane cultivar were tested, and another set of 35 buds were used as the control. The grown plants were randomly distributed in an intelligent artificial climate chamber with the abovementioned conditions. Three independent experiments were conducted. The disease severity and incidence of the leaf scald was recorded at 0-, 7-, 14-, 21-, and 28 dpi. 

### 3.6. Resistance Assessment of Sugarcane Cultivars against Leaf Scald

The disease index of sugarcane leaf scald disease was identified according to Rott et al. [43] and Zhao et al. [33]. At 28 dpi, the disease index (%) of all the tested sugarcane cultivars was calculated, and resistance grades were grouped according to the criteria of Fu et al. [44].

### 3.7. Combined Stress Treatments

Four sugarcane cultivars (ROC22, GT29, ZZ13, and LC09-15) were used for combined stress treatment. The buds were placed in a 32 °C incubator for germination for 3 days, and then cuttings with good bud germination were cultivated with clean water under the following conditions: temperature 30 °C, humidity 65%, and a light/dark cycle of 16/8 h. After the plants grew to the 3–5 leaf stage, they were cultured with Hoagland’s nutrient solution for 1 week, and then all the plants were divided into two groups for stress treatments. The first group was used for Xa-FJ1 inoculation without PEG6000 stress, while the second group was used for Xa-FJ1 inoculation plus 25% PEG6000 added in the Hoagland nutrient solution. Xa-FJ1 was inoculated using the leaf cutting method [45]. The treated plants were randomly distributed in an intelligent artificial climate chamber with the abovementioned conditions. Plant leaves were collected at 0 h and 24 hpt for the determination of subsequent bacterial contents, physiological and biochemical indexes, and gene expression. 

### 3.8. Determination of Pathogenic Bacterial Contents

Total DNA was extracted from sugarcane leaves with CTAB reagent for quantitative real-time PCR (qPCR) detection according to the specific protocol by Shi et al. [20]. At 0 and 24 hpt, population density determination was performed on three leaf subsamples collected for each treatment. Three technical replicates for each subsample were performed.

### 3.9. ROS Production and Antioxidant Enzyme Assays 

The ROS contents and activities of two antioxidant enzymes (SOD and CAT) were analyzed using the kit from Solarbio Science & Technology Co., Ltd. (Beijing, China) following the manufacturer’s manual. Briefly, 0.1 g of leaf samples was weighed and then ground with 1 mL of 10 mM PBS buffer (pH = 7.4), followed by centrifugation at 25 °C at 1200× *g* for 20 min, and then the supernatant was used for the subsequent determination of ROS contents by a sandwich ELISA, and the optical density at 450 nm was determined by spectrophotometry. Another set of leaf samples (0.1 g) were ground, followed by centrifugation (16,770× *g*) at 4 °C for 10 min. The supernatant was determined for CAT and SOD activity at visible wavelengths of 240 nm and 560 nm, respectively.

### 3.10. RNA Extraction and qRT-PCR Analysis 

The transcriptional expression of the genes was detected by a real-time quantitative reverse transcription PCR (qRT-PCR) assay with specific primer pairs (Appendix A). These genes include the respiratory burst oxidase homologs gene (*ScRBOHD*), which is critical in encoding ROS production in plants, and two genes encoding antioxidant enzymes, the superoxide dismutase gene (*ScSOD*) and the catalase gene (*ScCAT*). Total RNA was extracted from leaf subsamples, and they were reverse-transcribed into cDNA according to the method of Chu et al. [46]. For each treatment, three leaf subsamples were analyzed at each time point. Three technical replicates were carried out for each subsample. 

### 3.11. Statistical Analysis

Variance analysis (ANOVA) was utilized to compare the datasets. Duncan’s test (comparison among more than two groups of data) and Student’s *t*-test (comparison between two groups of data) were employed to determine mean differences at *p* < 0.05 or 0.01. Software from IBM (China), called SPSS version 18.0, was used for all the analyses. 

## 4. Conclusions

This study identified drought and leaf scald resistance in 16 recently released sugarcane cultivars, which offers a crucial hint for variety extension. Subsequently, four cultivars contrasting to drought and leaf scald resistance were treated with a combination of PEG6000 stress and *Xa* infection. Drought promoted the incidence of leaf scald disease and *Xa* contents in susceptible cultivars, while there was no significant change in resistant cultivars. The ROS burst and scavenging system was involved in four tested sugarcane cultivars against *Xa* infection. A stronger response of this pathway was observed in resistant cultivars than in susceptible cultivars. However, the response of the ROS production and scavenging system was weakened in sugarcane cultivars under combined stress (*Xa* infection along with PEG6000 stress) compared with *Xa* infection only. Notably, a higher weakening degree existed in susceptible cultivars than in resistant cultivars. Our findings suggest that ROS is a key defense node in sugarcane against *Xa* infection combined with drought stress. This work will lay the foundation for further research on the mechanism altering the prevalence and virulence of *Xa* in sugarcane under drought stress.

## Figures and Tables

**Figure 1 plants-13-00862-f001:**
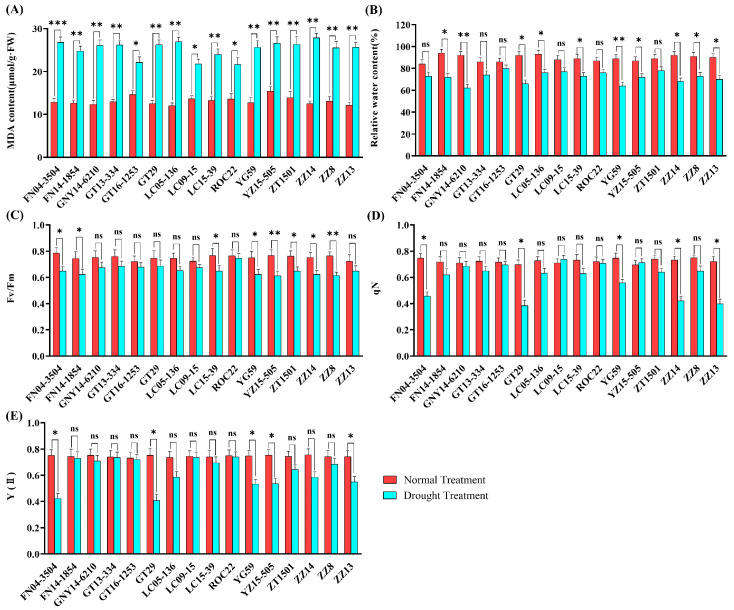
Changes in different physiological and biochemical indexes of 16 sugarcane cultivar leaves under normal and drought condition. (**A**) MDA contents, (**B**) relative water contents, (**C**) maximum quantum yield of PSII photochemistry (Fv/Fm), (**D**) non-photochemical quenching (qN), (**E**) PSII actual photosynthetic efficiency (Y(II)). Vertical bar values represent means ± SE. For each variety, significant differences between normal and drought treatments at *p* < 0.05, 0.01, and 0.001 (Student’s *t*-test) are indicated by one, two, and three asterisks, respectively. ns, not significant.

**Figure 2 plants-13-00862-f002:**
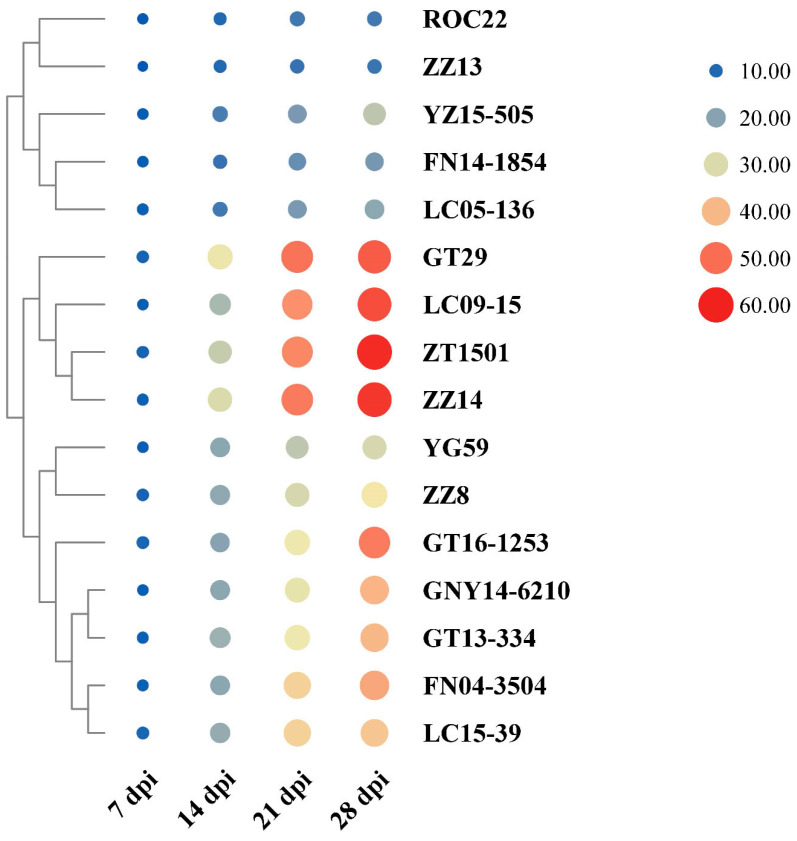
Disease index of 16 sugarcane cultivars under the infection by *X. albilineans* strain Xa-FJ1 at 7–28 days post-inoculation (dpi).

**Figure 3 plants-13-00862-f003:**
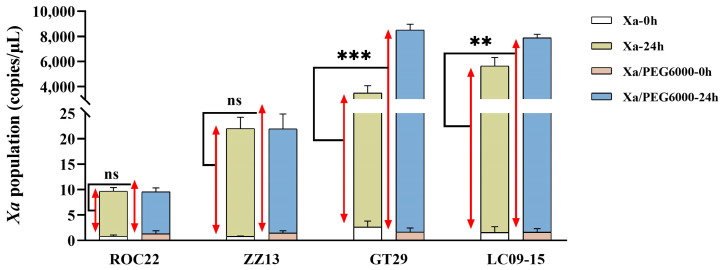
Quantitative PCR (qPCR) detection of bacterial pathogen population in four sugarcane cultivars under *X. albilineans* strain Xa-FJ1 infection or combing with PEG6000 stress. Xa-0h and Xa-24h: 0 and 24 h post-inoculation by Xa-FJ1, respectively; Xa/PEG6000-0h and Xa/PEG6000-24h: 0 and 24 h post-treatment with combined stress, respectively. ns, not significant; **, significant difference at *p* < 0.01; ***, significant difference at *p* < 0.001. Student’s *t*-test was used to determine mean differences. The red arrows indicate the bacterial pathogen populations in four sugarcane cultivars at 24 h after *X. albilineans* strain Xa-FJ1 infection alone (Xa-24h) or combined stress (Xa/PEG6000-24h).

**Figure 4 plants-13-00862-f004:**
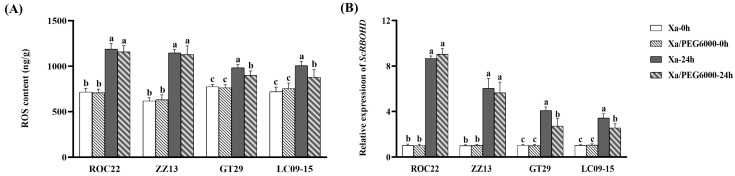
Changes in the ROS contents and transcriptional expression of *ScRBOHD* in sugarcane leaves under *X. albilineans* strain Xa-FJ1 infection along with PEG6000 stress. (**A**) ROS contents determined by an ELISA assay; (**B**) transcriptional expression of *ScRBOHD* determined by real-time quantitative reverse transcription PCR (qRT-PCR). Xa-0h and Xa-24h: 0 and 24 h post-inoculation by Xa-FJ1, respectively; Xa/PEG6000-0h and Xa/PEG6000-24h: 0 and 24 h post-treatment with Xa-FJ1 infection plus PEG6000 stress, respectively. Different letter between means indicates significant differences at *p* < 0.05 by Duncan’s test.

**Figure 5 plants-13-00862-f005:**
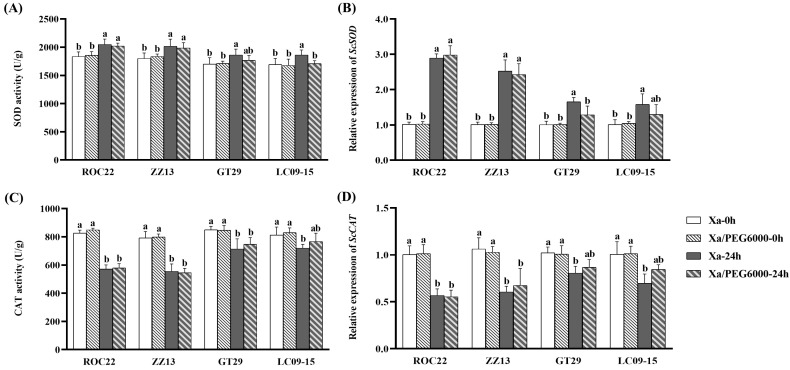
Changes in antioxidant enzyme activity and relative gene expression in sugarcane leaves *X. albilineans* infection along with PEG6000 stress. (**A**,**C**) Enzyme activities of SOD and CAT, respectively; (**B**,**D**) transcript levels of *ScSOD* and *ScCAT* determined by qRT-PCR, respectively. Xa-0h and Xa-24h: 0 and 24 h post-inoculation by Xa-FJ1, respectively; Xa/PEG6000-0h and Xa/PEG6000-24h: 0 and 24 h post-treatment with Xa-FJ1 infection plus PEG6000 stress, respectively. Different letter between means indicates significant differences at *p* < 0.05 by Duncan’s test. Same letters above bars indicates no significant difference between treatments.3. Discussion.

**Table 1 plants-13-00862-t001:** Membership function values and ranking of comprehensive evaluation values (D values) of 16 sugarcane cultivars.

Variety	Membership Function			D Value	Rank
	μ1	μ2	μ3		
LC09-15	0.9487	0.4645	0.4755	0.7764	1
GT16-1253	1.0000	0.3525	0.3186	0.7611	2
ROC22	0.9633	0.2351	0.5164	0.7498	3
GT13-334	0.6118	0.6142	0.5090	0.5945	4
LC15-39	0.5540	0.7773	0.4478	0.5775	5
FN14-1854	0.4478	0.8729	0.6594	0.5639	6
GNY14-6210	0.4030	0.6747	1.0000	0.5569	7
ZZ8	0.4036	1.0000	0.4619	0.5253	8
YZ15-505	0.4472	0.9969	0.2409	0.5145	9
ZT1501	0.5105	0.7291	0.2766	0.5110	10
LC05-136	0.3308	0.6219	0.4996	0.4144	11
YG59	0.1545	0.6518	0.5294	0.3123	12
ZZ13	0.1805	0.1863	0.4778	0.2329	13
ZZ14	0.0000	0.6090	0.5375	0.2068	14
FN04-3504	0.0847	0.4907	0.0000	0.1460	15
GT29	0.0173	0.0000	0.5373	0.1038	16

**Table 2 plants-13-00862-t002:** Different resistance grades of 16 sugarcane cultivars against *X. albilineans* strain Xa-FJ1 infection.

Grade	Variety (Line) Name	Number of Variety	Range of Disease Index (%)	Mean of Disease Index (%) ^a^
Resistant	ROC22, ZZ13	2	11.8–12.6	12.2 A
Medium resistant	FN14-1854, LC05-136, YZ15-505	3	18.1–26.3	21.7 B
Susceptible	YG59, ZZ8, FN04-3504, LC15-39, GNY14-6210, GT13-334, GT16-1253	7	29.4–48.4	38.9 C
High susceptible	GT29, ZZ14, LC09-15, ZT1501	4	52.8–58.8	55.8 D

^a^ Different letter between means indicates significant differences at *p* < 0.05 by Duncan’s test.

## Data Availability

All data supporting the findings of this study are available within the paper and its Appendix A.

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
