# Peer review of "Insights into Reactive Oxygen Species Production-Scavenging System Involved in Sugarcane Response to Xanthomonas albilineans Infection under Drought Stress"

_plants, 2024, doi:10.3390/plants13060862_

Round 1
Reviewer 1 Report
Comments and Suggestions for Authors
Dear Authors,
The subject of the study is interesting and topical, with high scientific and practical importance.
Some suggestions and corrections were made in the article.
The following aspects are brought to the attention of the authors.
1.
Citing bibliographic sources
Bibliographic sources are numbered in the order of citation in the text
Please check the bibliographic source [14], page 1, row 39, in the case of presenting the sources "[3,14]"
2.
Ionic forms
It is recommended to use the Equation Editor in Word to write the ionic forms correctly
e.g.
page 2, row 56
page 7, row 240
page 8, row 257
“(O2¯)” instead of current form
is "O-" and not "2-"
3.
The formula on page 8, row 283, could be presented as an equation, according to Instructions for Authors, and Microsoft Word template, Plants journal
Similarly, the relationships presented on page 9, row 302, and row 303 could also be considered.
This does not affect the scientific content and value of the article.
4.
Please check the bibliographic sources on page 9, rows 318 and 319
"Rott et al.", and "Zhao et al." indicate the same numbers of the bibliographic sources "[17,40]"
5.
References
According to Instructions for Authors, and Microsoft Word template, Plants journal
“Author 1, A.B.; Author 2, C.D. Title of the article. Abbreviated Journal Name Year, Volume, page range.”
“Include the digital object identifier (DOI) for all references where available.”
e.g.
page 11, rows 403 – 404
“Raymond, C.; Matthews, T.; Horton, R.M. The emergence of heat and humidity too severe for human tolerance. Sci. Adv. 2020, 6(19), eaaw1838. doi:10.1126/sciadv.aaw1838.”
Instead of
“Raymond, C.; Matthews, T.; Horton, R.M. The emergence of heat and humidity too severe for human tolerance. Science advances 2020, 6, eaaw1838, doi:10.1126/sciadv.aaw1838.”
Abbreviated Journal Name
Year
It is recommended to check the entire References chapter and make the necessary corrections

Author Response
Dear Authors,
The subject of the study is interesting and topical, with high scientific and practical importance.
Some suggestions and corrections were made in the article.
The following aspects are brought to the attention of the authors.
1.Citing bibliographic sources
Bibliographic sources are numbered in the order of citation in the text
Please check the bibliographic source [14], page 1, row 39, in the case of presenting the sources "[3,14]"
Author reply: The error has been fixed.
2.Ionic forms
It is recommended to use the Equation Editor in Word to write the ionic forms correctly
e.g.
page 2, row 56
page 7, row 240
page 8, row 257
“(O2¯)” instead of current form
is "O-" and not "2-"
Author reply: The error has been fixed.
3.The formula on page 8, row 283, could be presented as an equation, according to Instructions for Authors, and Microsoft Word template, Plants journal
Similarly, the relationships presented on page 9, row 302, and row 303 could also be considered.
This does not affect the scientific content and value of the article.
Author reply: The error has been fixed.
4.Please check the bibliographic sources on page 9, rows 318 and 319
"Rott et al.", and "Zhao et al." indicate the same numbers of the bibliographic sources "[17,40]"
Author reply: The error has been fixed.
5.References
According to Instructions for Authors, and Microsoft Word template, Plants journal
“Author 1, A.B.; Author 2, C.D. Title of the article. Abbreviated Journal Name Year, Volume, page range.”
“Include the digital object identifier (DOI) for all references where available.”
e.g.
page 11, rows 403 – 404
“Raymond, C.; Matthews, T.; Horton, R.M. The emergence of heat and humidity too severe for human tolerance. Sci. Adv. 2020, 6(19), eaaw1838. doi:10.1126/sciadv.aaw1838.”
Instead of “Raymond, C.; Matthews, T.; Horton, R.M. The emergence of heat and humidity too severe for human tolerance. Science advances 2020, 6, eaaw1838, doi:10.1126/sciadv.aaw1838.”
Abbreviated Journal Name
Year
It is recommended to check the entire References chapter and make the necessary corrections
Author reply: All references have been checked and modified.
Reviewer 2 Report
Comments and Suggestions for Authors
In this manuscript, Wei et al., presents a comprehensive study on the interaction between drought stress, Xanthomonas albilineans infection, and the reactive oxygen species (ROS) production-scavenging system in sugarcane.
Major:
#1. Why authors used different temperature settings in Xa inoculation and combined stress treatment? This variation may cause the variety selection step invalid.
#2. In plants, when ROS levels rise, the levels of CAT and SOD typically also increase. Here the authors notes a decrease in CAT levels across all four sugarcane varieties in response to rising ROS levels. The authors should also provide POD values to offer a more comprehensive understanding of the antioxidative response. Further, an explanation for the observed decrease in CAT levels is necessary.
#3. Why there were no single PEG6000 treatment groups for 4 variaties?
Minor:
#4. Xa-FJ1, the Xa should be italicized?
#5. Figure 5, second figure legend, 24h should be 0 h
Comments on the Quality of English LanguageEnglish can be improved.
e.g. Line 360. Three technical replicates 'were' carried out
Author Response
In this manuscript, Wei et al., presents a comprehensive study on the interaction between drought stress, Xanthomonas albilineans infection, and the reactive oxygen species (ROS) production-scavenging system in sugarcane.
Major:
#1. Why authors used different temperature settings in Xa inoculation and combined stress treatment? This variation may cause the variety selection step invalid.
Author reply: The 32°C and 30°C of the combined stress treatment section are the incubation temperatures for the sugarcane material. The experimental temperature for both the combined stress and Xa infection treatment was maintained at 28°C. The manuscript has been updated to include the conditions for the combined stress treatment.
#2. In plants, when ROS levels rise, the levels of CAT and SOD typically also increase. Here the authors notes a decrease in CAT levels across all four sugarcane varieties in response to rising ROS levels. The authors should also provide POD values to offer a more comprehensive understanding of the antioxidative response. Further, an explanation for the observed decrease in CAT levels is necessary.
Author reply: In our previous study, significant changes were observed in the levels of SOD and CAT following Xa infection, while there was no notable alteration in the levels of POD. Therefore, the measurement of POD content was not conducted in the present study. The relevant reference is:
[1] Zhao JY, Chen J, Shi Y, Fu HY, Huang MT, Rott PC, Gao SJ. Sugarcane responses to two strains of Xanthomonas albilineans differing in pathogenicity through a differential modulation of salicylic acid and reactive oxygen species. Front. Plant Sci. 2022 Dec 15;13:1087525. doi: 10.3389/fpls.2022.1087525.
Hence, POD content was not assessed in this study. Future comprehensive investigations will incorporate analysis of changes in POD content.
The explanation for the observed decrease in CAT levels has been added to the article.
#3. Why there were no single PEG6000 treatment groups for 4 variaties?
Author reply: Because this study was to explore the effect of drought stress on the incidence of sugarcane leaf scald disease, the premise of all studies was the incidence of leaf scald disease, so no separate PEG6000 osmotic stress treatment group was set up. Similar studies, such as:
[1] Qiu J, Liu Z, Xie J, Lan B, Shen Z, Shi H, Lin F, Shen X, Kou Y. Dual impact of ambient humidity on the virulence of Magnaporthe oryzae and basal resistance in rice. Plant Cell Environ. 2022 Dec;45(12):3399-3411.
[2] Qiu J, Xie J, Chen Y, Shen Z, Shi H, Naqvi NI, Qian Q, Liang Y, Kou Y. Warm temperature compromises JA-regulated basal resistance to enhance Magnaporthe oryzae infection in rice. Mol. Plant. 2022 Apr 4;15(4):723-739.
Minor:
#4. Xa-FJ1, the Xa should be italicized?
Author reply: Italics are not needed here
#5. Figure 5, second figure legend, 24h should be 0 h
Author reply: The error has been fixed.
English can be improved.
Author reply: English has improved.
e.g. Line 360. Three technical replicates 'were' carried
Author reply: The error has been fixed.
Reviewer 3 Report
Comments and Suggestions for Authors
Insights into Reactive Oxygen Species Production-scavenging System Involved in Sugarcane Response to Xanthomonas albilineans Infection under Drought Stress
In this study, the drought and leaf scald resistance were identified on 16 sugarcane cultivars using Xanthomonas albilineans (Xa) inoculation and soil drought treatments, respectively. Then, four cultivars contrasting to drought and leaf scald resistance were used for exploring the mechanisms of drought affecting Xa-sugarcane interaction. Drought stress significantly increased the occurrence of leaf scald and Xa contents in susceptible varieties but had no obvious effect on resistant varieties. The ROS bursting and scavenging system was significantly activated in sugarcane in the process of Xa infection, particularly in the resistant varieties. Compared with Xa infection alone, defense response via the ROS generating and scavenging system was obviously weakened in sugarcane (especially in susceptible varieties) under Xa infection plus drought stress. The authors concluded that, ROS might play a crucial role involving in sugarcane defense against combined effects of Xa infection and drought stress.
The MS is clearly written but some issues should be considered to improve the quality of the manuscript. The comments and suggestions are as follows:
- Line 32. (have shown that that extreme) Please delete the repeated word.
- Line 74. from leaves of 16 sugarcane materials. Are they materials or cultivars?
- The same in Line 75.
- Lines 191-197. Please revise and re-edit this part.
- Lines 203-204. Please revise the sentence. It is not clear
- I noticed that all results presented in Fig. 1 are not discussed in discussion part. Please explain.
- The authors at least should write something about the relation between drought and RWC and MDA which support their results. For example: (Many reports have shown that when plants are exposed to drought stress, their leaves exhibit great reductions in their RWC and considerable increase in MDA (Hessini et al., 2022; Mazrou et al., 2023)
Hessini, K.; Wasli, H.; Al-Yasi, H.M.; Ali, E.F.; Issa, A.A.; Hassan, F.A.S.; Siddique, K.H.M. Graded Moisture Deficit Effect on Secondary Metabolites, Antioxidant, and Inhibitory Enzyme Activities in Leaf Extracts of Rosa damascena Mill. var. trigentipetala. Horticulturae 2022, 8, 177. https://doi.org/10.3390/horticulturae8020177.
Mazrou, R.M.; Hassan, F.A.S.; Mansour, M.M.F.; Moussa, M.M. Melatonin Enhanced Drought Stress Tolerance and Productivity of Pelargonium graveolens L. (Herit) by Regulating Physiological and Biochemical Responses. Horticulturae 2023, 9, 1222. https://doi.org/10.3390/horticulturae9111222
Lines 249-253. The observation demonstrated that the rice mutant osnramp1 (natural resistance-associated macrophage proteins) can maintain ROS homeostasis by upregulating SOD activity and decreasing CAT activity, which ultimately helps plants acquire broad-spectrum resistance to bacterial and fungal pathogens [34]. Please revise and re-edit the sentence.
- Generally, the style and quality of English language in discussion part is lower than that in the introduction part.
- Line 268. What do you mean by flowing top-water?
- Lines 269-270. Where the cuttings were grown until the stage of 3–5 leaves? Inside or outside the incubation chamber? And how long?
- Line 272. What is the area of the incubation chamber? And please provide the light intensity inside.
- Lines 278-280. The authors mentioned that Leaf samples were collected at 0 and 3 days after drought treatment for the subsequent determination of drought resistance indicators. This is my major concern. Is 3 days enough to apply drought treatment on sugarcane? Is preventing irrigation for 3 days consider drought stress? Please explain.
- Line 329. The authors used 25% PEG6000 to apply drought stress treatment in combined stress treatment. Please explain why the authors did not use the same treatment for applying the drought stress treatment alone. This is critical comment.
- The experimental design is missed. Is it RD? or factorial or what?
- Line 341. The title 4.9 Physico-biochemical assays. Please modify it. For example: ROS production and antioxidant enzyme assays.
-
-
Comments on the Quality of English Language
Minor editing is required
Author Response
In this study, the drought and leaf scald resistance were identified on 16 sugarcane cultivars using Xanthomonas albilineans (Xa) inoculation and soil drought treatments, respectively. Then, four cultivars contrasting to drought and leaf scald resistance were used for exploring the mechanisms of drought affecting Xa-sugarcane interaction. Drought stress significantly increased the occurrence of leaf scald and Xa contents in susceptible varieties but had no obvious effect on resistant varieties. The ROS bursting and scavenging system was significantly activated in sugarcane in the process of Xa infection, particularly in the resistant varieties. Compared with Xa infection alone, defense response via the ROS generating and scavenging system was obviously weakened in sugarcane (especially in susceptible varieties) under Xa infection plus drought stress. The authors concluded that, ROS might play a crucial role involving in sugarcane defense against combined effects of Xa infection and drought stress.
The MS is clearly written but some issues should be considered to improve the quality of the manuscript. The comments and suggestions are as follows:
- Line 32. (have shown that that extreme) Please delete the repeated word.
Author reply: The error has been fixed.
- Line 74. from leaves of 16 sugarcane materials. Are they materials or cultivars?
Author reply: It should be "cultivars". The error has been fixed.
- The same in Line 75.
Author reply: It should be "cultivars". The error has been fixed.
- Lines 191-197. Please revise and re-edit this part.
Author reply: This part has been revised and re-edited.
- Lines 203-204. Please revise the sentence. It is not clear
Author reply: This sentence has been revised and re-edited.
- I noticed that all results presented in Fig. 1 are not discussed in discussion part. Please explain.
Author reply: Since this part of the experiment aims to assess the drought resistance level of 16 sugarcane varieties using drought resistance indicators rather than investigating changes in physiological and biochemical indexes of sugarcane after drought stress, Fig. 1 was not included in the discussion.
- The authors at least should write something about the relation between drought and RWC and MDA which support their results. For example: (Many reports have shown that when plants are exposed to drought stress, their leaves exhibit great reductions in their RWC and considerable increase in MDA (Hessini et al., 2022; Mazrou et al., 2023)
Hessini, K.; Wasli, H.; Al-Yasi, H.M.; Ali, E.F.; Issa, A.A.; Hassan, F.A.S.; Siddique, K.H.M. Graded Moisture Deficit Effect on Secondary Metabolites, Antioxidant, and Inhibitory Enzyme Activities in Leaf Extracts of Rosa damascena Mill. var. trigentipetala. Horticulturae 2022, 8, 177. https://doi.org/10.3390/horticulturae8020177.
Mazrou, R.M.; Hassan, F.A.S.; Mansour, M.M.F.; Moussa, M.M. Melatonin Enhanced Drought Stress Tolerance and Productivity of Pelargonium graveolens L. (Herit) by Regulating Physiological and Biochemical Responses. Horticulturae 2023, 9, 1222. https://doi.org/10.3390/horticulturae9111222
Author reply: The relationship and references have been added to the text.
Lines 249-253. The observation demonstrated that the rice mutant osnramp1 (natural resistance-associated macrophage proteins) can maintain ROS homeostasis by upregulating SOD activity and decreasing CAT activity, which ultimately helps plants acquire broad-spectrum resistance to bacterial and fungal pathogens [34]. Please revise and re-edit the sentence.
Author reply: This sentence has been revised and re-edited.
- Generally, the style and quality of English language in discussion part is lower than that in the introduction part.
Author reply: The language quality of the discussion section has been improved.
- Line 268. What do you mean by flowing top-water?
Author reply: The error has been fixed. It should be ” flowing tap-water”.
- Lines 269-270. Where the cuttings were grown until the stage of 3–5 leaves? Inside or outside the incubation chamber? And how long?
Author reply: The planting environment, management methods, and time required for sugarcane growth have been added to the text.
- Line 272. What is the area of the incubation chamber? And please provide the light intensity inside.
Author reply: The manufacturer, model, and light intensity of the intelligent artificial climate chamber have been added to the text.
- Lines 278-280. The authors mentioned that Leaf samples were collected at 0 and 3 days after drought treatment for the subsequent determination of drought resistance indicators. This is my major concern. Is 3 days enough to apply drought treatment on sugarcane? Is preventing irrigation for 3 days consider drought stress? Please explain.
Author reply: By the time we stopped watering for three days, the soil moisture content had reached the range of moderate drought stress, i.e., 60%–70%. Therefore, in the experiment, the sampling time was taken as a three-day stop for watering.
- Line 329. The authors used 25% PEG6000 to apply drought stress treatment in combined stress treatment. Please explain why the authors did not use the same treatment for applying the drought stress treatment alone. This is critical comment.
Author reply: Compared with the gradual soil drying methods, the simulated drought using PEG6000 osmotic stress is more responsive, and the experimental environment is more controllable and stable. This accelerated process allows us to more easily and reliably study the molecular mechanisms of stress combinations including drought stress. After obtaining certain basic research results, we will consider using gradual soil drying methods for validation and further research.
- The experimental design is missed. Is it RD? or factorial or what?
Author reply: The experiment was conducted using the Randomized Complete Block Design (RCBD) for allocation.The experimental design has been added to the article.
- Line 341. The title 4.9 Physico-biochemical assays. Please modify it. For example: ROS production and antioxidant enzyme assays.
Author reply: The title 4.9 has been modified.
Round 2
Reviewer 2 Report
Comments and Suggestions for Authors
The manuscript has been improved.
Line 274 'Unexpectedly, our results showed that the levels of ROS and SOD activity in susceptible cultivars exposed to PEG6000 stress combined with Xa infection were markedly higher than those under Xa infection alone. This phenomenon needs to be explained in further work'. Why it was unexpectedly? This explanation is super confusing. And this is also the reason of including a PEG6000 group here.
Author Response
Comment: The manuscript has been improved.
Response: Esteemed reviewer, thank you for your appreciation. We believe that the corrections will meet with the approval.
Comment: Line 274 'Unexpectedly, our results showed that the levels of ROS and SOD activity in susceptible cultivars exposed to PEG6000 stress combined with Xa infection were markedly higher than those under Xa infection alone. This phenomenon needs to be explained in further work'. Why it was unexpectedly? This explanation is super confusing. And this is also the reason of including a PEG6000 group here.
Response: Thank you for highlighting the mistake. Yes you are right that under drought stress ROS accumulation increased to a greater extent as evidenced from previous literature. We have corrected the sentence and added the reason that ROS production might be increased due to the drought stress. Further, we have also added recent reference to support our statement.
Reviewer 3 Report
Comments and Suggestions for Authors
The authors carefully revised the MS and addressed all of my comments. The MS can be accepted in current form.
Comments on the Quality of English LanguageMinor editing is required
Author Response
Comment: The authors carefully revised the MS and addressed all of my comments. The MS can be accepted in current form.
Response: Esteemed reviewer, thank you for your appreciation. We believe that the corrections will meet with the approval.
Comment: Minor editing is required
Response: We have thoroughly checked the manuscript and tried our best to remove any minor linguistic mistake.
Round 3
Reviewer 2 Report
Comments and Suggestions for Authors
The manuscript has been improved.